# ADVERSARIAL LATENT REPRESENTATION FOR POSITIVE UNLABELED LEARNING

## ABSTRACT

Novelty detection, a widely studied problem in machine learning, is the task of detecting a novel class of data that has not been previously observed. Deep networks have driven the state-of-the-art work on this application in recent years due to their successful applications on large and more complex datasets. The usual setting for novelty detection is unsupervised whereby only examples of the normal class are available during training, but more recently there has been a surge in interest in semi-supervised methods. A common assumption about semi-supervised methods is their access to an additional set of labeled data that includes a few examples of anomalies. Transductive novelty detection or positive-unlabeled (PU) learning on the other hand assumes access to an additional unlabeled set that contains examples of anomalies. In this study, we focus on machine vision applications and propose TransductGAN, a transductive generative adversarial network (GAN) that attempts to learn how to generate image examples from the novel class by separating the latter from the negative class in a latent space using a mixture of two Gaussians. It achieves that by incorporating an adversarial autoencoder with a GAN network; the ability to generate examples of novel data points offers not only a visual representation of the new class, but also overcomes the hurdle faced by many inductive methods about how to tune the model hyperparameters at the decision rule level. In addition, the introduction of a latent space enables an enhanced discriminative learning. Our model has shown superior performance over state-of-the-art work on several benchmark datasets.

## 1 INTRODUCTION

Novelty detection has been addressed widely in the machine learning literature, its basic form aims to build a decision rule in order to distinguish a normal set of data points/ inliers from novelties/ anomalies. Due to their superior ability to handle high dimensional data, deep networks have overtaken shallow methods such as one-class SVM Schölkopf et al. (1999); Tax & Duin (2004) in recent years in driving the state-of-the-art (SOTA) work in applications such as machine vision. A common setting that deep network applications follow in novelty detection is unsupervised whereby only examples of the normal data are available during training. Some notable contributions include deep one class classification Ruff et al. (2018), adversarial models that rely on reconstructions errors Akcay et al. (2019); Schlegl et al. (2017); Zenati et al. (2018a); Akçay et al. (2019); Zenati et al. (2018b); Perera et al. (2019), deep energy-based models Zhai et al. (2016), autoencoding Gaussian mixture models Zong et al. (2018), perturbation-based learning models Cai & Fan (2022) and one-class anomaly classifiers using adversarially interpolated training samples Chen et al. (2022).

But there could be applications where an additional contaminated dataset is also available during training, incorporating this set into the learning is referred to as semi-supervised. One subbranch of semi-supervised learning assumes access to a labeled contaminated dataset. Most of the work using that assumption has focused on shallow methods such as Görnitz et al. (2013) or is application or data-specific such as Ergen & Kozat (2020); Min et al. (2018); Pang et al. (2019). Deep SAD Ruff et al. (2020) is a more general method that was introduced as an extension to the unsupervised deep one class classification Ruff et al. (2018) to account for the semi-supervised case.

Another slightly more challenging semi-supervised subbranch assumes access to an unlabelled contaminated dataset. The latter goes by the name of transductive novelty detection Blanchard et al.

(2010) or PU learning. With the emergence of adversarial training, GAN-based Goodfellow et al. (2014) models have become state-of-the-art in PU learning applications and mostly involve learning a binary classifier from positive and unlabelled data. In Chiaroni et al. (2018) a positive GAN (PGAN) learns to generate counter-examples by being trained to learn the distribution of the unlabelled dataset. It exploits the weakness in GAN training in order to achieve that but its performance is limited to cases where the generated distribution does not perfectly match the unlabelled data distribution. In Hou et al. (2018) a generative positive-unlabeled (GenPU) model is proposed that makes use of a series of discriminators and generators to produce both positive and negative samples; GenPU requires prior knowledge of the contamination rate in the unlabeled set. Following the latter two methods, Chiaroni *et al.* Chiaroni et al. (2020) have proposed another two-stage GAN model (D-GAN) that learns the counter-examples distribution. D-GAN incorporates a biased PU risk in the discriminator loss function that constrains the generator to learn the positive samples distribution exclusively. The study also demonstrates that the standard GAN loss function in use also alleviates the need for prior knowledge of the contamination rate. Concurrently Chen et al. (2020) proposes a variational approach and Zhang et al. (2020) uses a mixture model of restricted Boltzmann machines that also surpasses the need for a prior estimation.

Indeed, the latter study proposes to minimize the overlap between the normal class conditional PDF and the anomaly class conditional PDF by separating them. In this study we adopt a similar idea but propose to achieve that using a lower-dimensional latent space where separating the two PDFs is more efficient. To the best of our knowledge, none of the SOTA work on PU learning has attempted to explore learning in a latent space using deep networks. Our proposed TransductGAN model fills that gap precisely, it combines an adversarial autoencoder (Makhzani et al., 2016) that maps the data to a latent space with a GAN. It attempts to simultaneously separate the inliers from the novelties in the latent space and learn how to generate anomalies. A standard binary classifier can then be trained using the artificially generated novel and normal samples projections in order to perform anomaly detection. A diagram of TransductGAN is provided in Figure 1 and our main contributions are summarized below:

- we propose a novel GAN-based model capable of learning how to generate novelties in a transductive/ PU learning setup

- by being able to generate the anomalies artificially we can train a binary classifier and tune its hyperparameters without explicitly accessing the unknown class; this overcomes the hurdle faced by many unsupervised methods of how to tune their hyperparameters in the absence of novel data examples

- the introduction of a latent space constitutes our main contribution; it enables the use of more discriminative features by modeling two non-overlapping Gaussian distributions in the latent space

- we thoroughly test our model on three benchmark datasets using two different protocols

- our model exhibits superior performance over several SOTA unsupervised and semi-supervised models

## 2 TRANSDUCTGAN

### 2.1 MODEL OBJECTIVES

In the below we will refer to the normal/ novel class as positive/ negative class respectively. TransductGAN's objectives are to ensure the distributions of these two classes remain non-overlapping in the latent space while it learns to generate examples of the negative class. The transductive setup assumes access to a positive dataset and an unlabeled dataset containing examples of the negative class. TransductGAN includes an adversarial autoencoder Makhzani et al. (2016) that matches the aggregated posterior distribution of the latent space with a bimodal distribution (a mixture of two Gaussians), and a generative model that learns how to map the negative class mode in the latent space to image examples belonging to the negative class. A binary classifier (e.g. a support vector machine) can then be trained to identify the novel data points in the unlabeled set. This operation takes place in the absence of any labeled negative data examples.

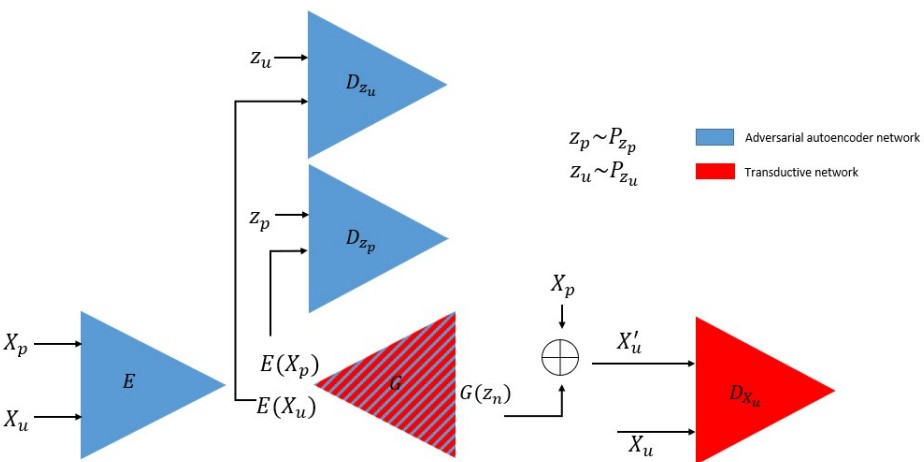

Figure 1: TransductGAN. The network includes an adversarial autoencoder that ensures the latent distributions match with the priors as per equation 5, and a transductive network that learns to map samples from $p_n(\mathbf{z})$ into images that belong to the novel class.

More formally, let us define the input as $x \in \mathbb{R}^m$ and its projection in the latent space (i.e. the output of the encoder $E$ in Figure 1) as $z \in \mathbb{R}^n$ with $m >> n$. We have access to the positive dataset $X_p = \{x_p^i\}_{i=1}^{n_p} \sim P_{x_p}$ and an unlabeled dataset that includes novel data $X_u = \{x_u^i\}_{i=1}^{n_u} \sim P_{x_u}$. Let us also define $y \in \{-1, +1\}$ as the class label (negative or positive). The class conditional negative density (representing the negative samples in the latent space) is:

$$q_n(z) = q(z|y = -1), \tag{1}$$

and the class conditional positive density (representing the positive data samples) in the latent space:

$$q_p(z) = q(z|y = 1). \tag{2}$$

The unlabeled density in the latent space is hence defined as:

$$q_u(z) = \pi q_n(z) + (1 - \pi)q_p(z). \tag{3}$$

$\pi$ is the prior for the negative class also known as the contamination rate, we assume we have access to this figure and that it is strictly positive $\pi > 0$:

$$\pi = p(y = -1) \tag{4}$$

The prior distributions we want to impose directly are $p_p(z)$ for the positive dataset with $N(\mu_p, \Sigma_p)$ and $p_u(z)$ for the unlabeled dataset, and indirectly $p_n(z)$ for the negative dataset with $N(\mu_n, \Sigma_n)$. The unlabeled prior can also be defined as:

$$p_u(z) = \pi p_n(z) + (1 - \pi)p_p(z) \tag{5}$$

We assume we have access to $n_p$ positive samples and $n_u$ unlabeled samples.

## 2.2 TRAINING METHODS

The proposed model for learning how to generate negative and positive samples is summarised in Figure 1. The first part of the model is an adversarial autoencoder that includes the networks $E$, $G$, and the critic networks $D_{z_u}$ and $D_{z_p}$. $E$ and $G$ are trained to minimize a reconstruction loss, or a minimal $L_2$ distance between an image and its reconstructed version:

$$L_{reconstruction} = \min_E \min_G \mathbb{E}_{x \sim P_x} \|x - G(E(x))\|_2 \tag{6}$$

In the above $P_x$ could refer to either $P_{X_u}$ or $P_{X_p}$.

When using $X_u$, $D_{z_u}$ and $E$ are trained to match the projections of $X_u$ onto the latent space with $p_u(z)$ defined in equation 5, a Wasserstein loss with a gradient penalty (WGAN-GP) (Gulrajani et al., 2017) is used in order to ensure that. And similarly when using $X_p$, $D_{z_p}$ and $E$ are trained to match the projections of $X_p$ onto the latent space with $p_p(z)$. In both cases, the latent loss can be defined as:

$$L_{regularization} = \min_E \max_D \mathbb{E}_{x \sim P_x}[D(E(x))] - \mathbb{E}_{z \sim p(z)} D(z) + \lambda \mathbb{E}_{\hat{z} \sim P_{\hat{z}}}[(\|\nabla_{\hat{z}} D(\hat{z})\|_2 - 1)^2] \quad (7)$$

In the above, depending on whether we are using $X_u$ or $X_p$ during training, $P_x$ refers to either $P_{X_u}$ or $P_{X_p}$. $p(z)$ refers to either $p_u(z)$ or $p_p(z)$ and the critic network $D$ refers to either $D_{z_u}$ or $D_{z_p}$. Similarly to the gradient penalty definition in (Gulrajani et al., 2017), the sampling $P_{\hat{z}}$ is a uniform distribution along straight lines between pairs of points sampled from $(q_u(z), p_u(z))$ or $(q_p(z), p_p(z))$. Crucially, by imposing $p_u(z)$ and $p_p(z)$ on $q_u(z)$ and $q_p(z)$ respectively, $q_n(z)$ will implicitly match with $p_n(z)$.

The second part of the model focuses on $G$ and $D_{X_u}$, the generator $G$ will be trained to produce fake negative samples $G(z_n)$. Let us define $X_u'$ to be the result of a concatenation operation between $X_p$ and $G(z_n)$ with proportions of $1 - \pi$ and $\pi$ respectively. With regards to generating $G(z_n)$, the intuition is that $X_u$ is formed by samples from both the positive set $X_n$ and unknown negative samples $X_n$, we ensure $X_p$ already forms a subset of $X_u'$, the generator $G$ is then left with the task of complementing $X_u'$ with samples not already provided by $X_p$, i.e. samples that are similar to the novel samples $X_n$. The role of $D_{x_u}$ is hence to challenge $G$ to produce a set $G(z_n)$ that is indistinguishable from $X_n$.

The adversarial loss, also a WGAN-GP, is defined as:

$$L_{adv} = \min_G \max_{D_{X_u}} \mathbb{E}[D_{X_u}(X_u')] - \mathbb{E}[D_{X_u}(X_u)] + \lambda \mathbb{E}_{\hat{x} \sim P_{\hat{x}}}[(\|\nabla_{\hat{x}} D_{X_u}(\hat{x})\|_2 - 1)^2] \quad (8)$$

Similarly to the gradient penalty definition in (Gulrajani et al., 2017), the sampling $P_{\hat{x}}$ is a uniform distribution along straight lines between pairs of points sampled from $(X_u', X_u)$. A pseudocode of TransductGAN is provided in algorithm 0.

Once the TransductGAN training is completed, a binary classifier can be trained using the latent projections of $X_p$ onto the latent space as one class and the latent projections of the fake negative images as the other class as per Algorithm 2.

## 3 EXPERIMENTS

### 3.1 DATASETS

#### 3.1.1 MNIST LECUN & CORTES (2010).

This dataset consists of 28x28 grayscale images of handwritten digits split evenly across 10 classes. The training set includes 60000 examples and the test set includes 10000 examples.

#### 3.1.2 FASHION-MNIST XIAO ET AL. (2017)

This dataset shares the same specifications as MNIST in terms of size and train/ test split, it consists of grayscale images of different fashion products.

#### 3.1.3 CIFAR10 KRIZHEVSKY & HINTON (2009).

This dataset consists of 32x32 coloured images of animals or modes of transport with 50000 training samples and 10000 test samples. The dataset is split evenly across 10 classes.

---

**Algorithm 1** TransductGAN. We use default values of $\lambda = 10$, $n_{critic} = 5$ (Gulrajani et al., 2017).

---

**Require:** the contamination rate $\pi$, the gradient penalty coefficient $\lambda$, the number of critic iterations per generator iteration $n_{critic}$, the batch size $m$.
**Require:** initial critics parameters $\theta_{D_{z_u}}$, $\theta_{D_{z_p}}$, $\theta_{D_{X_u}}$, initial encoder parameters $\theta_E$ and initial generator parameters $\theta_G$.
    select next batch of $m$ samples from $X_u$
    minimize $L_{rec}$ wrt $\theta_E$ and $\theta_G$ and update parameters accordingly
    sample $z \sim p_u(z)$
    minimize $L_{reg}$ wrt $\theta_E$ and update parameters accordingly
    **for** $i = 1, ..., n_{critic}$ **do**
        maximize $L_{reg}$ wrt $\theta_{D_{z_u}}$ and update parameters accordingly
    **end for**
    select next batch of $m$ samples from $X_p$
    minimize $L_{rec}$ wrt $\theta_E$ and $\theta_G$ and update parameters accordingly
    sample $z \sim p_p(z)$
    minimize $L_{reg}$ wrt $\theta_E$ and update parameters accordingly
    **for** $i = 1, ..., n_{critic}$ **do**
        maximize $L_{reg}$ wrt $\theta_{D_{z_p}}$ and update parameters accordingly
    **end for**
    select next batch of $m$ samples from $X_u$
    sample $int(\pi * m)$ samples $z \sim p_n(z)$
    combine $G(z)$ with $(m - int(\pi * m))$ samples from $X_p$ to form $X_u'$
    minimize $L_{adv}$ wrt $\theta_G$ and update parameters accordingly
    **for** $i = 1, ..., n_{critic}$ **do**
        maximize $L_{adv}$ wrt $\theta_{D_{X_u}}$ and update parameters accordingly
    **end for**

---

---

**Algorithm 2** Binary classifier for novelty detection.

---

**Require:** Training of TransductGAN as per Algorithm 0
    sample 5000 samples $z_n$ from $p_n(z)$
    select 5000 samples $X_p$ from $X_n$
    train a two-class SVM with linear kernel with $E(G(z_n))$ as one class and $E(X_p)$ as another
    apply classifier on $E(X_u)$ as novelty detector

---

## 3.2 Protocols

We will use two widely used protocols when running our experiments. In the first protocol, we will treat one of the classes as novel and remove it from the training set and the rest of the classes will be treated as normal. In the second protocol we will adopt the one-versus-all approach where a single class will be treated as normal and the rest of the classes as anomalies. In both protocols we will iterate through all the class combinations when reporting our results.

## 3.3 Performance measure

Our performance is measured using the area under the curve of the Receiver Operating Characteristic (AUROC). The Receiver Operating Characteristic curve plots the true positive rate against the false positive rate as we vary the threshold of our classifier. We also include image outputs of the produced novel images.

## 3.4 Network architectures

The transductive network architecture as highlighted in red in Figure 1 follows the same implementation [1] as was provided by Gulrajani et al. (2017). Our open source implementation will provide further details about the adversarial network implementation.

## 3.5 Methods for comparison

When using protocol 1, we compare our model against the unsupervised shallow OCSVM model Schölkopf et al. (1999); Tax & Duin (2004), the unsupervised deep EGBAD Zenati et al. (2018a) and GANomaly Akcay et al. (2019) models and a variant of the semi-supervised D-GAN model Chiaroni et al. (2020). We use the scikit implementation of OCSVM [2] with the 'scale' default kernel width value. We follow the same training procedures stated in the original publications regarding EGBAD and GANomaly's results. EGBAD's results with CIFAR10 were taken from (Akcay et al., 2019). The D-GAN implementation made available by the authors did not include the version they used with MNIST and CIFAR10 datasets; our attempt to reproduce their results resulted in a mode collapse with classification results no better than random so we have not included these in our comparison. We have however built a variant of their model which we call D-GAN-V. It adopts an architecture that is identical to D-GAN with the exception that we are now using the earth mover distance as the loss function; this alleviates the problem of mode collapse but because of this modification the model now requires prior knowledge of the contamination rate. In addition, the generator and discriminator networks are identical to the ones used in TransductGAN, this ensures a fair comparison to D-GAN.

When using protocol 2 we compare our results against three unsupervised methods with deep networks, namely one-class deep SVDD Ruff et al. (2018) (DSVDD), deep structured energy-based model (DSEBM) Zhai et al. (2016) and one-class GAN (OCGAN) Perera et al. (2019). For the case of semi-supervised learning competitors we choose a method that have made use of labeled anomalies and report results from Ruff et al. (2020) on deep SAD, in this case we choose the setting with no polluted data in the training set and use a 10% ratio of labeled anomalies. We also include the results of the PU learning method D-GAN from Chiaroni et al. (2020).

## 3.6 Crucial model parameters

We have highlighted previously that the main contribution of this work is the introduction of a latent space where we train the data projections to fit a bimodal distribution. In order to ensure the distributions of the positive and negative data projections do not overlap in the latent space it is essential to set the means and covariances of the priors carefully.

---

[1]https://github.com/igul222/improved_wgan_training
[2]https://scikit-learn.org/stable/modules/generated/sklearn.svm.OneClassSVM.html

## 3.7 RESULTS

We present our results for protocol 1 in Tables 1, 5 and 4 for MNIST, CIFAR10 and Fashion-MNIST respectively. The results are broken down by rows corresponding to the normal data (included in the leftmost column). The best average results are highlighted in bold. Due to time constraints we only include the results for D-GAN-V and our method with Fashion-MNIST. For the same protocol, we also include some output examples in 2 and notice the presence of normal data points within the fake negative dataset. We include further image outputs in the appendix while varying the contamination rate. Unsurprisingly the generated examples are less polluted with a higher contamination rate. For protocol 2 we only report the average results only for all three datasets.

Our method outperforms its unsupervised learning counterparts by the largest margins, this is unsurprising given that we make use of the anomalies present in the test set albeit without the labels. TransductGAN also outperforms its PU learning competitors D-GAN/ D-GAN-V by a smaller margin than the unsupervised methods. This outperformance highlights the advantage of introducing a latent space. In fact we go further and conduct an ablation study where we compare our model against a vanilla model where the adversarial network is completely omitted (the adversarial network is coloured in blue in Figure 1. The vanilla network does not make use of a latent space and trains a binary classifier (a SVM with a radial basis function kernel) based on the fake (negative) generated samples from the generator and the positive real images $X_p$, it is summarised in Figure 3) and an outline of the overall procedure is provided in Algorithm 3 and Algorithm 4 in the appendix. As we can clearly see from Table 5, the latent space offers 11.1%, 9.6%, and 8.2% improvements over its vanilla counterpart for 5%, 10% and 30% contamination rate respectively. This suggests that a latent space is all the more relevant as the examples of anomalies become rarer in the unlabelled set. Finally, TransductGAN also outperforms deepSAD although the latter makes use of a few labeled anomalies. TransductGAN does carry an advantage over deepSAD in protocol2 where it is able to see a large amount of anomalous data (the contamination rate is 90%).

|   | OCSVM | GANomaly | EGBAD | TransductGAN | D-GAN-V |
|---|---|---|---|---|---|
| 0 | 0.853(0) | 0.882 | 0.86 | 0.991(0.002) | 0.995(0) |
| 1 | 0.315(0.001) | 0.663 | 0.314 | 0.996(0) | 0.996(0) |
| 2 | 0.775(0.002) | 0.952 | 0.835 | 0.985(0.002) | 0.981(0.001) |
| 3 | 0.655(0.004) | 0.794 | 0.712 | 0.975(0.005) | 0.972(0.004) |
| 4 | 0.498(0.003) | 0.803 | 0.655 | 0.978(0.004) | 0.976(0.005) |
| 5 | 0.589(0.003) | 0.864 | 0.713 | 0.975(0.002) | 0.962(0.004) |
| 6 | 0.691(0.003) | 0.852 | 0.753 | 0.994(0.002) | 0.994(0) |
| 7 | 0.582(0.009) | 0.697 | 0.525 | 0.976(0.005) | 0.974(0.001) |
| 8 | 0.544(0.002) | 0.792 | 0.728 | 0.967(0.001) | 0.964(0.003) |
| 9 | 0.349(0.005) | 0.534 | 0.547 | 0.968(0.005) | 0.912(0.063) |
|   | 0.585 | 0.783 | 0.664 | **0.98** | 0.973 |

Table 1: ROC (AUC) results summary for protocol 1 - MNIST. Three different random seeds are used with standard deviations shown in brackets.

## 4 CONCLUSION AND FUTURE WORK

We have introduced a latent representation for PU/ transductive learning methods that take advantage of a lower-dimensional space in order to separate the normal data from the anomalies. Our method outperforms several SOTA unsupervised and semi-supervised learning methods. One of the main disadvantage of our method is its need to have access to the contamination rate, we plan to work on estimating that figure in future work. We have also noticed through the results in Table 5 that our model performs better with a higher contamination rate, this may in fact be a weakness in the model (and more generally in PU learning algorithms) and we leave it for future studies to ensure the model's performance does not degrade as the examples of anomalies become rarer in the unlabeled set. Finally we also plan to incorporate self-supervised learning in our work following the example of Golan & El-Yaniv (2018) in order to enhance the feature extraction in the latent space.

|       | OCSVM         | GANomaly | EGBAD | TransductGAN   | DGAN-V        |
|-------|---------------|----------|-------|----------------|---------------|
| plane | 0.54(0.01)    | 0.622    | 0.582 | 0.81(0.004)    | 0.786(0.009)  |
| car   | 0.657(0.002)  | 0.632    | 0.527 | 0.85(0.002)    | 0.77(0.007)   |
| bird  | 0.384(0.011)  | 0.513    | 0.386 | 0.66(0.007)    | 0.633(0.015)  |
| cat   | 0.577(0.012)  | 0.575    | 0.455 | 0.655(0.004)   | 0.64(0)       |
| deer  | 0.32(0.005)   | 0.591    | 0.385 | 0.686(0.007)   | 0.62(0.008)   |
| dog   | 0.57(0.005)   | 0.625    | 0.490 | 0.715(0.011)   | 0.68(0.018)   |
| frog  | 0.397(0.001)  | 0.668    | 0.359 | 0.736(0.015)   | 0.643(0.007)  |
| horse | 0.54(0.005)   | 0.650    | 0.527 | 0.74(0.011)    | 0.696(0.012)  |
| ship  | 0.51(0.009)   | 0.622    | 0.411 | 0.87(0.006)    | 0.836(0.019)  |
| truck | 0.66(0.006)   | 0.615    | 0.554 | 0.81(0.009)    | 0.776(0.005)  |
|       | 0.515         | 0.611    | 0.468 | **0.7533**     | 0.7053        |

Table 2: ROC (AUC) results summary for protocol 1 - CIFAR10. Three different random seeds are used with standard deviations shown in brackets.

|             | D-GAN-V       | TransductGAN   |
|-------------|---------------|----------------|
| T-shirt/top | 0.868(0.002)  | 0.955(0.005)   |
| Trouser     | 0.922(0.002)  | 0.996(0.000)   |
| Pullover    | 0.872(0.006)  | 0.954(0.005)   |
| Dress       | 0.893(0.000)  | 0.974(0.002)   |
| Coat        | 0.880(0.003)  | 0.951(0.005)   |
| Sandal      | 0.763(0.002)  | 0.993(0.001)   |
| Shirt       | 0.848(0.005)  | 0.891(0.01)    |
| Sneaker     | 0.928(0.000)  | 0.993(0.000)   |
| Bag         | 0.917(0.001)  | 0.991(0.000)   |
| Ankle boot  | 0.921(0.003)  | 0.995(0.000)   |
|             | 0.881         | **0.969**      |

Table 3: ROC (AUC) results summary - Fashion-MNIST. Three different random seeds are used with standard deviations shown in brackets.

|               | DSVDD  | DSEBM | OCGAN  | deepSAD | D-GAN | TransductGAN |
|---------------|--------|-------|--------|---------|-------|--------------|
| MNIST         | 0.948  | -     | 0.9750 | 0.97    | 0.989 | **0.995**    |
| Fashion-MNIST | -      | 0.866 | -      | 0.91    | -     | **0.983**    |
| CIFAR10       | 0.6481 | 0.609 | 0.6566 | 0.8     | 0.815 | **0.841**    |

Table 4: ROC (AUC) results summary for protocol 2.

|       | TransductGAN(5%) | Vanilla(5%)   | TransductGAN(10%) | Vanilla(10%)  | TransductGAN(30%) | Vanilla(30%)  |
|-------|------------------|---------------|-------------------|---------------|-------------------|---------------|
| plane | 0.763(0.003)     | 0.625(0.005)  | 0.81(0.004)       | 0.786(0.009)  | 0.86(0.002)       | 0.84(0)       |
| car   | 0.7(0.009)       | 0.756(0.018)  | 0.85(0.002)       | 0.77(0.007)   | 0.906(0.004)      | 0.843(0.037)  |
| bird  | 0.6(0.018)       | 0.39(0.011)   | 0.66(0.007)       | 0.633(0.015)  | 0.766(0.008)      | 0.713(0.018)  |
| cat   | 0.586(0.045)     | 0.56(0.013)   | 0.655(0.004)      | 0.64(0)       | 0.786(0.009)      | 0.69(0.025)   |
| deer  | 0.55(0.007)      | 0.376(0.055)  | 0.686(0.007)      | 0.62(0.008)   | 0.796(0)          | 0.713(0.006)  |
| dog   | 0.65(0.021)      | 0.62(0.01)    | 0.715(0.011)      | 0.68(0.018)   | 0.816(0.006)      | 0.753(0.004)  |
| frog  | 0.613(0.022)     | 0.493(0.073)  | 0.736(0.015)      | 0.643(0.007)  | 0.86(0.011)       | 0.746(0.011)  |
| horse | 0.636(0.021)     | 0.606(0.007)  | 0.74(0.011)       | 0.696(0.012)  | 0.856(0.009)      | 0.78(0.031)   |
| ship  | 0.8(0.007)       | 0.736(0.008)  | 0.87(0.006)       | 0.836(0.019)  | 0.9(0.001)        | 0.89(0.003)   |
| truck | 0.696(0.027)     | 0.77(0.007)   | 0.81(0.009)       | 0.776(0.005)  | 0.876(0.005)      | 0.816(0.028)  |
|       | 0.6596           | 0.5935        | 0.7533            | 0.7053        | 0.8426            | 0.7786        |

Table 5: ROC (AUC) results summary for ablation study - CIFAR10. Results for TransductGAN and the vanilla model for a varying contamination rate in brackets. Three different random seeds are used with standard deviations included (using protocole 1).

### ACKNOWLEDGMENTS

Use unnumbered third level headings for the acknowledgments. All acknowledgments, including those to funding agencies, go at the end of the paper.

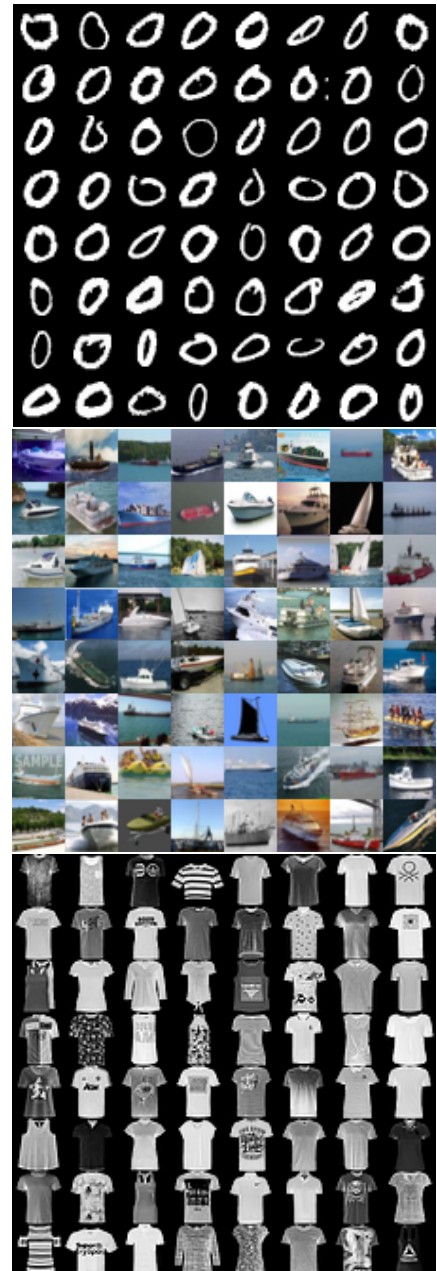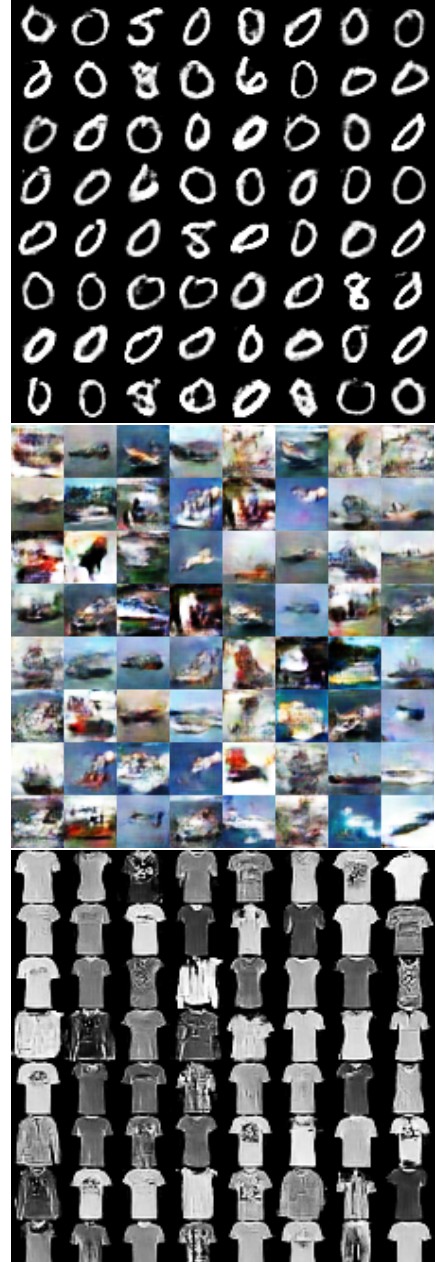

Figure 2: Examples of generated outputs. The left column represents real images and the right column represents examples of generated anomalies using protocol 1. The anomaly categories are '0', 'ship' and 'T-shirt/top' for the respective datasets.

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

## A  APPENDIX

### A.1  VANILLA MODEL

In this section we include details related to the vanilla model.

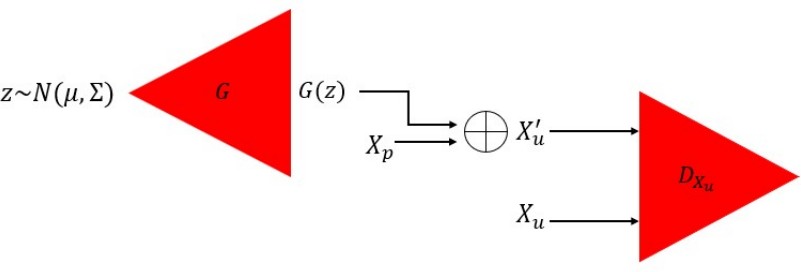

Figure 3: Vanilla model.

---

**Algorithm 3** Vanilla model. We use default values of $\lambda = 10$, $n_{critic} = 5$ (Gulrajani et al., 2017).

---

**Require:** the contamination rate $\pi$, the gradient penalty coefficient $\lambda$, the number of critic iterations per generator iteration $n_{critic}$, the batch size $m$.
**Require:** initial critics parameters $\theta_{D_{X_u}}$ and initial generator parameters $\theta_G$.
    select next batch of $m$ samples from $X_u$
    sample $int(\pi * m)$ samples $z \sim N(\mu, \Sigma)$
    combine $G(z)$ with $(m - int(\pi * m))$ samples from $X_p$ to form $X'_u$
    minimize $L_{adv}$ wrt $\theta_G$ and update parameters accordingly
    **for** $i = 1, ..., n_{critic}$ **do**
        maximize $L_{adv}$ wrt $\theta_{D_{X_u}}$ and update parameters accordingly
    **end for**

---

**Algorithm 4** Binary classifier for novelty detection with vanilla model.

---

**Require:** Training of vanilla model as per Algorithm 3
    sample 5000 samples $z$ from $N(\mu, \Sigma)$
    select next batch of 5000 samples from $X_p$
    train a two-class SVM with radial basis function kernel with $G(z)$ as one class and $X_p$ as another
    apply classifier on $E(X_u)$ as novelty detector

---

### A.2  EFFECT OF CONTAMINATION RATE

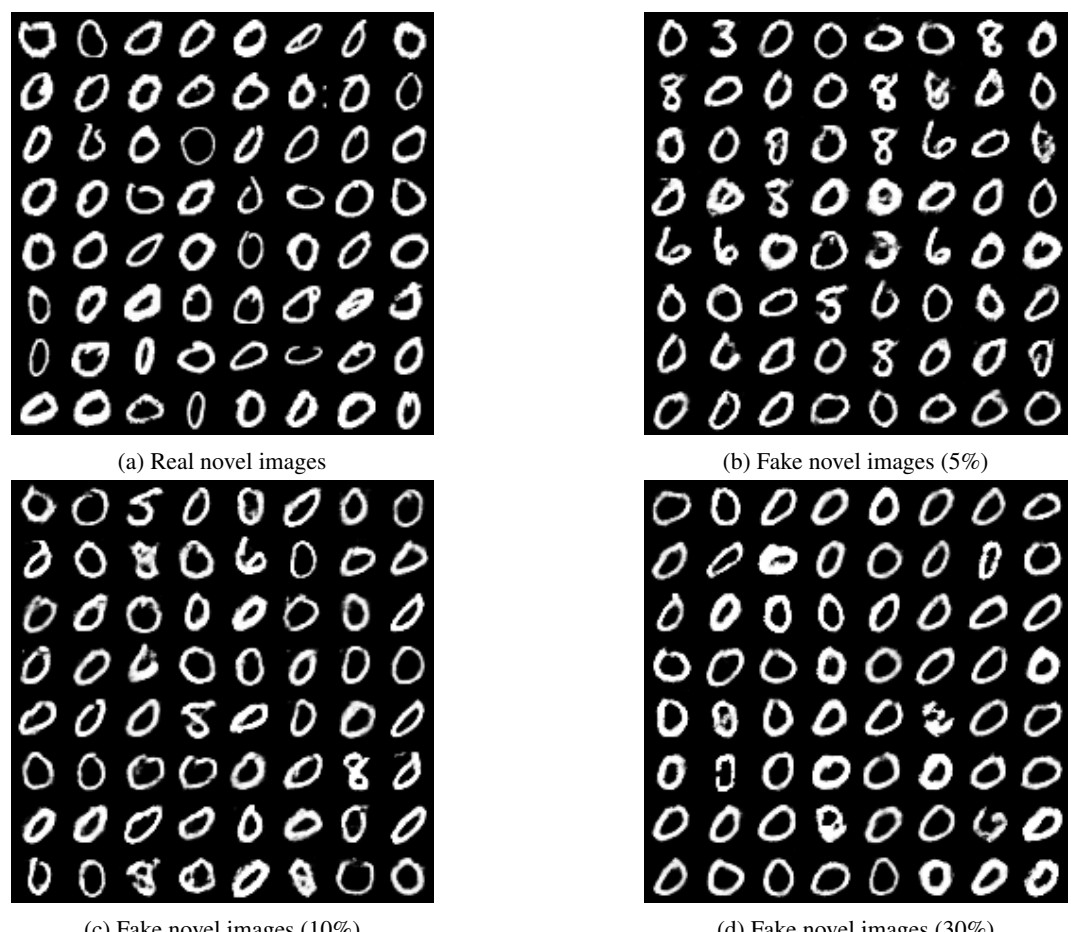

(a) Real novel images

(b) Fake novel images (5%)

(c) Fake novel images (10%)

(d) Fake novel images (30%)

Figure 4: MNIST example with '0' as novel class. For the fake examples, the value in brackets corresponds to the contamination rate in the test set that was used during training.

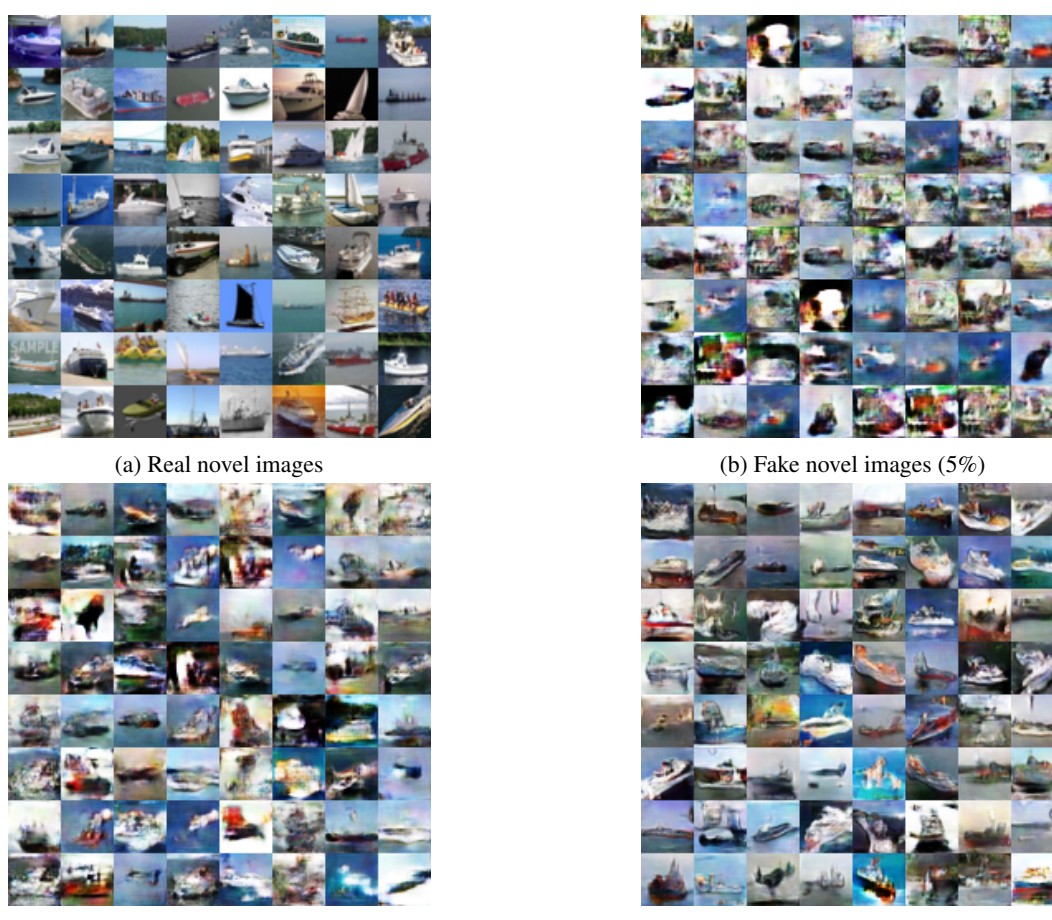

(a) Real novel images

(b) Fake novel images (5%)

(c) Fake novel images (10%)

(d) Fake novel images (30%)

Figure 5: CIFAR10 example with ship as novel class. For the fake examples, the value in brackets corresponds to the contamination rate in the test set that was used during training.

