# OpenReview forum: "Adversarial latent representation for positive unlabeled learning"
_ICLR.cc/2024/Conference — Submitted to ICLR 2024_

### Official Review · Reviewer_JuuS · 2023-10-31

**Soundness:** 2 fair
**Presentation:** 1 poor
**Contribution:** 1 poor
**Rating:** 3
**Confidence:** 3

**Summary:**

This paper used a GAN model to generate images from a novel class by formulating the novelty detection problem as a PU learning problem.
The proposed method was tested on semi-synthetic datasets.

**Strengths:**

- Formulating the novelty/anomaly detection problem as PU learning might be a good idea

**Weaknesses:**

- Due to the unclear presentation, I was not able to fully understand the proposed method. Several symbols were not clearly defined and explained. I had to guess the meanings of many parts. For example:
  - $P_{x_p}$ and $P_{x_u}$ were not defined and the notation is ambiguous.
  - $N$ is Gaussian distribution?
  - $L$ defined in this way is a value, not a learning objective.
- The Gaussian mixture model may not fully capture the posterior distribution.
- The figure and algorithm are hard to follow.
- The experimental setting may not be a good proxy of real-world novelty detection problems.

**Questions:**

- I cannot understand the meaning of "how to tune the model hyperparameters at the decision rule model".
- "is used to ensure that": ensure what?

---

### Official Review · Reviewer_NiHf · 2023-10-31

**Soundness:** 2 fair
**Presentation:** 2 fair
**Contribution:** 2 fair
**Rating:** 3
**Confidence:** 4

**Summary:**

This paper focused on machine vision applications and proposed Transductive GAN, a generative adversarial network (GAN) that attempts to learn how to generate image examples from the novel class.

**Strengths:**

1. The written is clear and easy to follow.

**Weaknesses:**

1. As a GAN model, the experiments should generally be applied to a larger dataset, .e.g., ImageNet. The evaluation metric of AUC is too limited. FID like metrics should also be considered.
2. This paper used a well established model to solve the novelty detection task. The overall contribution of this work is limited, since the GAN related machine learning methodologies used in the paper have been addressed in many previous works. This work may fit a data conference better.

**Questions:**

The assumption of a latent space using a mixture of two Gaussians seems too coarse to me. In general, either the unknown class or the known class may include many sub-classes with various features or latent representations. What if the data has complex taxonomy, how two Gaussian distributions can capture the data structure?

---

### Official Review · Reviewer_ozUy · 2023-11-01

**Soundness:** 2 fair
**Presentation:** 1 poor
**Contribution:** 1 poor
**Rating:** 3
**Confidence:** 4

**Summary:**

This paper focus on semi-supervised novelty detection and its machine vision applications, introducing an approach named TransductGAN. This method employs a transductive generative adversarial network (GAN) to generate image examples from previously unseen classes by separating them from negative examples in a latent space using a mixture of two Gaussians. By combining an adversarial autoencoder with a GAN network, the approach not only provides visual representations of new classes but also addresses challenges related to model hyperparameter tuning in inductive methods. In addition, the incorporation of a latent space enhances discriminative learning.

**Strengths:**

1. It is interesting to generate examples of novel data points by incorporating an adversarial autoencoder with a GAN network.

**Weaknesses:**

1. The description of some concepts is confusing, such as the relationship among novelty detection, semi-supervised learning and positive-unlabeled learning.
2. The methodology section lacks clarity and detail, making it challenging to understand the entire process.
3. More recent studies related to this work should be discussed and compared.
4. There are some significant issues with the overall writing of this paper, such as some citation format error and the content at the bottom of page 8.
5. The missing data in Table 4 should be filled in, and the experimental results have not been thoroughly analyzed.

**Questions:**

1. The description of the task and corresponding concepts should be reorganized.
2. Provide a more detailed and logically structured explanation of the methodology.
3. The content of the entire paper requires comprehensive revision and enhancement.

---

### Official Review · Reviewer_DxcQ · 2023-11-01

**Soundness:** 3 good
**Presentation:** 3 good
**Contribution:** 2 fair
**Rating:** 6
**Confidence:** 4

**Summary:**

The paper proposes a novel model called TransductGAN for transductive generative adversarial network (GAN) in positive-unlabeled (PU) learning. It  aims to separate the positive class (novel data) from the negative class (normal data) in a latent space using a mixture of two Gaussians. It combines an adversarial autoencoder with a GAN network to generate examples of novel data points. The proposed model is evaluated on benchmark datasets and shows superior performance compared to state-of-the-art unsupervised and semi-supervised models.

**Strengths:**

(1) The paper presents an innovative approach in transductive PU learning by incorporating a latent space and using a GAN-based model.
(2) The method is well-motivated and addresses the challenge of tuning hyperparameters in unsupervised methods without access to novel data examples.
(3) The paper provides a thorough evaluation of the proposed model on benchmark datasets and demonstrates its superior performance over existing models.

**Weaknesses:**

(1) The paper lacks implementation details, making it difficult to reproduce the study.
(2) The evaluation could benefit from more ablation studies and comparisons with additional baselines.
(3) The paper could provide more clarity in the exposition to improve reader understanding.

**Questions:**

1. Can you provide more implementation details, such as the specific architecture used for the networks and the hyperparameter settings?
2. Can you include additional ablation studies to further analyze the performance of the proposed model?
3. Have you considered comparing your method with more baselines in the field to provide a more comprehensive evaluation?

---

### Meta-Review · Area_Chair_3zyt · 2023-12-19

**Metareview:**

The paper proposes a GAN model with a mixture of two Gaussians as latent distribution for the task of positive unlabeled learning.

The idea of applying a GAN to positive unlabeled learning is novel and interesting.

However the paper could be significantly improved by improving writing, ensuring reproducibility, and extending the empirical evaluation with ablation studies, comparisons with additional baselines, experiments on larger data sets and additional evaluation metrics like FID.

**Justification For Why Not Higher Score:**

In the current form the paper does not guarantee reproducibility and does not provide a thorough  empirical evaluation.

**Justification For Why Not Lower Score:**

N/A

---

### Decision · Program_Chairs · 2024-01-16

Reject